# The Role of Micro Breaking of Small-Scale Wind Waves in Radar Backscattering from Sea Surface

**Irina A. Sergievskaya [1,\*], Stanislav A. Ermakov [1,2], Aleksey V. Ermoshkin [1], Ivan A. Kapustin [1], Olga V. Shomina [1] and Alexander V. Kupaev [1]**

[1]  Institute of Applied Physics, Russian Academy of Sciences, 603950 Nizhny Novgorod, Russia; stas.ermakov@ipfran.ru (S.A.E.); eav@ipfran.ru (A.V.E.); kia@ipfran.ru (I.A.K.); shomina@ipfran.ru (O.V.S.); kupaev@ipfran.ru (A.V.K.)

[2]  Volga State University of Water Transport, 603950 Nizhny Novgorod, Russia

[\*]  Correspondence: i.sergia@ipfran.ru; Tel.: +7-831-416-4935

**Abstract:** The study of the microwave scattering mechanisms of the sea surface is extremely important for the development of radar sensing methods. Some time ago, Bragg (resonance) scattering of electromagnetic waves from the sea surface was proposed as the main mechanism of radar backscattering at moderate incidence angles of microwaves. However, it has been recently confirmed that Bragg scattering is often unable to correctly explain observational data and that some other physical mechanisms should be taken into consideration. The newly introduced additional scattering mechanism was characterized as non-polarized, or non-Bragg scattering, from quasi-specular facets appearing due to breaking wave crests, the latter usually occurring in moderate and strong winds. In this paper, it was determined experimentally that such non-polarized radar backscattering appeared not only for rough sea conditions in which wave crests strongly break and "white caps" occur, but also at very low wind velocities close to their threshold values for the wave generation process. The experiments were performed using two polarized Doppler radars. The experiments demonstrated that a polarization ratio, which characterizes relative contributions of non-polarized and Bragg components to the total backscatter, changed slightly with wind velocity and wind direction. Detailed analysis of radar Doppler shifts revealed two types of scatterers responsible for the non-polarized component. One type of scatterer, moving with the velocities of decimeter-scale wind waves, determined radar backscattering at low winds. We identified these scatterers as "microbreakers" and related them to nonlinear features in the profile of decimeter-scale waves, like bulges, toes and parasitic capillary ripples. The scatterers of the second type were associated with strong breaking, moved with the phase velocities of meter-scale breaking waves and appeared at moderate winds additionally to the "microbreakers". Along with strong breakers, the impact of microbreaking in non-polarized backscattering at moderate winds remained significant; specifically the microbreakers were found to be responsible for about half of the non-polarized component of the radar return. The presence of surfactant films on the sea surface led to a significant suppression of the small-scale non-Bragg scattering and practically did not change the non-Bragg scatterer speed. This effect was explained by the fact that the nonlinear structures associated with dm-scale waves were strongly reduced in the presence of a film due to the cascade mechanism, even if the reduction of the amplitude of dm waves was weak. At the same time, the velocities of non-Bragg scatterers remained practically the same as in non-slick areas since the phase velocity of dm waves was not affected by the film.

**Keywords:** dual-polarized radars; Bragg and non-Bragg mechanism; polarization ratio; Doppler shifts; microbreaking; strong wavebreaking

## 1. Introduction

At present, microwave radar, in particular the C- and X-bands, is a widely used remote sensing tool for sea state monitoring, providing high-resolution imagery of both spatial and temporal variations of the sea surface [1,2]. Accordingly, the investigation of mechanisms of radar backscattering from the sea surface is extremely important for the development of remote sensing methods [3–6]. The increased interest in this issue in the context of ocean remote sensing is associated, on the one hand, with the intensification of ocean exploration and, on the other hand, with the appearance of new instruments, including multiband Doppler radars operating simultaneously at several polarizations (see [7–11] and references therein).A Bragg scattering mechanism was previously considered as the principal mechanism of radar backscattering at moderate incidence angles [3]. However, new data obtained using advanced techniques, in particular co-polarized microwave radar, have indicated some limitations of the Bragg mechanism associated with breaking of wind waves. It was suggested that these limitations could be removed by taking into consideration non-Bragg, so-called non-polarized, scattering [6,12,13]. This non-polarized scattering was hypothesized to be significant at moderate incidence angles and at moderate and strong winds (6–11 m/s) when strong wavebreaking occurs [12,13]. In [14–16] it was suggested that along with the wave breaking process, at low grazing angles sharp-crested waves significantly contribute to the non-Bragg radar return. According to a semi-empirical model of normalized radar cross-section (NRCS) proposed in [6,12,13], the non-Bragg mechanism is associated with specular tilted facets distributed in the profiles of breaking waves. However, since these facets have scales comparable with microwave wavelengths, it is more correct to interpret non-Bragg scattering as diffraction on small-scale quasi-specular facets on the wavy surface with randomly distributed breaking zones—that is, as overturning wave crests [17]. The experiments in [18,19] confirmed that the appearance of small-scale structures ("bulges" and "toes" [20–22]) characterized by high curvature values in the profiles of short gravity waves of cm–dm–scales resulted in an increase of the non-Bragg component. Detailed schematics of the nonlinearity in the profile of dm-scale waves are presented in Figure 1 of [20]. As a parameter characterizing the contribution of the non-Bragg component in the total radar backscatter, a polarization ratio defined as a ratio of intensities at VV and HH polarizations is usually used (see, for example, [17,23,24] and references therein).It was shown in [10] that intense spikes in the radar return occurred dat moderate wind velocities and non-Bragg backscattering contributed significantly in radar return both at VV and HH polarizations in the zones visually associated with overturning wave crests. Outside the spikes, the polarization ratio increased, sometimes up to the values corresponding to the Bragg mechanism. This indicated that the intensity of non-Bragg scattering varied considerably along the dominant wind wave profile.

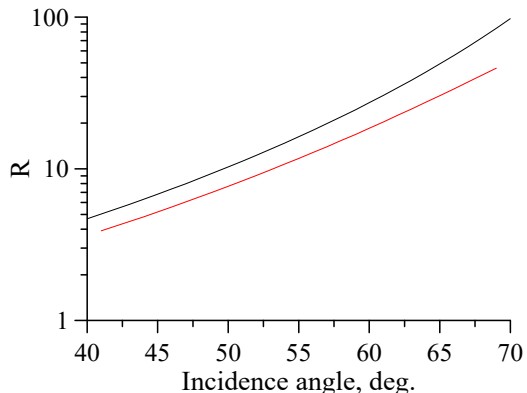

**Figure 1.** R (black curve) vs. the local incidence angle. Red curve—$\langle R(\theta) \rangle$.

The primary focus of this paper is on the nature of non-Bragg effects produced by local nonlinearities in the profiles of short wind waves on the ocean surface in a wide wind velocity range as revealed by the analysis of data from field experiments on radar remote sensing using two dual (VV/HH)-polarized

microwave Doppler radars working at moderate incident angles. The paper is organized as follows. Section 2 is devoted to the theoretical background that provides the basis for investigating the role of microbreaking in the formation of a non-Bragg component. Section 3 describes field experiments recently performed on an oceanographic platform in the Black Sea. The results of our field experiments are collected in Section 4 and discussed in Section 5. The conclusions are presented in Section 6.

## 2. Theoretical Background and Data Processing

According to [15], the total instantaneous normalized radar cross-section (NRCS) at VV and HH polarizations $\sigma^0_{VV,HH}$ is supposed to be a sum of Bragg and non-Bragg parts:

$$\sigma^0_{VV,HH} = \sigma^0_{B\_VV,HH} + \sigma^0_{NBC} \tag{1}$$

The Bragg component can be written as:

$$\sigma^0_{BC\_VV,HH} = 16\pi k_e^4 g^2_{VV,HH}(\theta)F(\vec{k}_B), \tag{2}$$

where $F(\vec{k}_B)$ is a spectrum of wind waves at a Bragg wave number $k_B = 2k_e sin\theta$, $\theta$ is the incidence angle, $k_e$ is the wave number of incident electromagnetic waves and $g^2_{VV,HH}$ is the backscattering Fresnel coefficient depending on the polarizations of incident/reflected electromagnetic waves and on the incidence angle [3]. The non-Bragg component in Equation (1) is assumed to be the same both at VV and HH polarizations [15]. One can thus acquire the instantaneous polarization difference (PD), the non-Bragg component (NBC) and the polarization ratio (PR) as follows:

$$PD = \sigma^0_{B\_VV} - \sigma^0_{B\_HH} = 16\pi k_e^4 (g^2_{VV}(\theta) - g^2_{HH}(\theta))F(\vec{k}_B), \tag{3}$$

$$NBC = \sigma^0_{NBC} = \frac{\sigma^0_{VV} - R\sigma^0_{HH}}{1 - R}, \tag{4}$$

$$PR = \sigma^0_{VV}/\sigma^0_{HH}. \tag{5}$$

Here $R(\theta) = \frac{g^2_{VV}(\theta)}{g^2_{HH}(\theta)}$ is the polarization ratio from Bragg scattering theory.

The polarization difference (Equation (3)) characterizes Bragg scattering and is determined by the wind wave spectrum at the Bragg wavenumber and by the difference of the backscattering coefficients. One should note that the Bragg component (or PD) depends on the local incidence angle $\theta = \theta_0 - \zeta'_x(r,t)$ and, since the latter varies in the presence of long surface waves, on the long wave slopes $\zeta'_x(r,t)$.

At low winds (<6 m/s) the long wave slopes are small, so one can neglect the variations of the local incident angle and consider $R(\theta)$ as a constant $R(\theta_0)$. At large enough long wave slopes, one has to take into account the variations of the coefficients $g^2_{VV,HH}$ due to long waves. When considering the NBC averaged over the periods of dominant surface waves, one can easily obtain the equation for the NBC-averaged value, which can be written as:

$$NBC = \langle\sigma^0_{NBC}\rangle = \frac{\langle\sigma^0_{VV}\rangle - \langle R\rangle\langle\sigma^0_{HH}\rangle}{1 - \langle R\rangle} \tag{6}$$

here $\langle R(\theta)\rangle = \frac{\langle g^2_{VV}(\theta)F(\vec{k}_B)\rangle}{\langle g^2_{HH}(\theta)F(\vec{k}_B)\rangle}$, where the brackets $\langle \ldots \rangle$ denote averaging over scales much larger than the periods of dominant surface waves. The polarization ratio of the pre-averaged intensities is $\overline{PR} = \langle\sigma^0_{VV}\rangle/\langle\sigma^0_{HH}\rangle$. Figure 1 shows $R(\theta)$ as a function of the local incidence angle for X-band Bragg waves; the red curve denotes $\langle R(\theta)\rangle$ for the case of the long wave slope variance in the observation direction of about 0.01 [16]. It can be seen that $R(\theta)$ strongly depends on the incidence angle and therefore can significantly vary due to variations of the local incidence angle along the dominant wave

profile. For instance, 95% of the surface slopes for the Gaussian distribution are within two wave slope variances, so that the polarization ratio at a mean incidence angle $\theta_0 \approx 60°$ varies between five and several tens of units. Hence, in order to correctly estimate the local values of the non-Bragg component, it is necessary to retrieve the local tilts due to the long waves.

Variations of the local incidence angle due to long waves can be estimated from the fluctuations of the Doppler shift of the polarization difference spectrum. The output signal I(t) for a coherent radar can be written as (see, e.g., [25]):

$$I(t) = Re\left[A(t)e^{i\varphi(t)}\right],\tag{7}$$

where $A(t)$ and $\varphi(t) = 2\vec{k}_e\vec{r}(t)$ are the current amplitude and phase of the signal, $\vec{k}_e$ is the wave vector of the incident electromagnetic wave and $\vec{r}(t)$ is the radius vector of the surface scattering element. An instantaneous spectrum of radar return is:

$$S(\omega,t) = \frac{1}{2\pi}\overline{\int Re\left[A(t)e^{i\varphi(t)}\right]Re\left[A(t+\tau)e^{i\varphi(t+\tau)}\right]e^{i\omega\tau}d\tau},\tag{8}$$

where $\omega$ is the frequency of the spectral component. The output signal amplitude is assumed to slowly vary on time scales of the order of the inverse frequencies of the radar Doppler spectrum. Then, it follows from Equation (8) that:

$$S(\omega,t) = \frac{1}{2}P(t)[\delta(\omega - 2k_e V) + \delta(\omega + 2k_e V)],\tag{9}$$

where $V = \frac{dr(t)}{dt}$ is the speed of microwave scatterers and $P(t)$ is the backscattering intensity.

The instantaneous radar Doppler shift $F_D$ can be written as:

$$F_D = 1/2\pi\cdot\vec{k}_B[\vec{V}_s + \vec{U}_{orb}(t)].\tag{10}$$

Here $\vec{V}_s$ is a sum of the velocities of scatterers and of a steady surface current and $\vec{U}_{orb}(t)$ is the orbital velocity of dominant surface waves. Mean velocities of Bragg/non-Bragg scatterers can be obtained by averaging the instantaneous offset of the Doppler spectrum of Bragg/non-Bragg components if the current velocity is known. The Doppler shift variations are related to the orbital velocities of long waves for, e.g., the case of upwind observation, as follows:

$$\delta F_D(t) = \frac{1}{\pi}k_e(U(t)sin\theta - W(t)cos\theta).\tag{11}$$

where $U(t)$ and $W(t) = \frac{dh(t)}{dt}$ are the horizontal and vertical components of the orbital velocity of the dominant waves and $h(t)$ is the elevation of the surface. For deep water waves, the amplitudes of the horizontal and vertical velocities are equal and the velocities are shifted by $\frac{\pi}{2}$ relative to each other. To obtain the local tilts of long waves $\eta(t)$, we assume that the elevations in long waves can be represented as Fourier series:

$$h(t) = \sum_n \frac{\eta_{sn}}{K(\Omega_n)}sin(\Omega_n t) + \frac{\eta_{cn}}{K(\Omega_n)}cos(\Omega_n t),\tag{12}$$

Here $\Omega_n = \frac{2\pi n}{T}$, $T$ is the time interval of data analysis, $\frac{\eta_{sn}}{K(\Omega_n)}$ and $\frac{\eta_{cn}}{K(\Omega_n)}$ are the Fourier coefficients, $K(\Omega_n) = \Omega_n^2/g$ and $g$ is the gravitational acceleration. Then, the slopes of long waves and vertical and horizontal velocities are represented as:

$$\eta(t) = \sum_n \eta_{cn}sin(\Omega_n t) - \eta_{sn}cos(\Omega_n t),\tag{13}$$

$$W(t) = \frac{dh(t)}{dt} = \sum_n \frac{\eta_{sn}\Omega_n}{K(\Omega_n)}cos(\Omega_n t) - \frac{\eta_{cn}\Omega_n}{K(\Omega_n)}sin(\Omega_n t), \tag{14}$$

$$U(t) = \sum_n \frac{\eta_{sn}\Omega_n}{K(\Omega_n)}sin(\Omega_n t) - \frac{\eta_{cn}\Omega_n}{K(\Omega_n)}cos(\Omega_n t). \tag{15}$$

After substituting the expressions for $W(t)$ and $U(t)$ in Equation (11), one obtains:

$$F_D(t) = \frac{1}{\pi}k_e \sum_n F_{sn}sin(\Omega_n t) + F_{cn}cos(\Omega_n t), \tag{16}$$

where

$$F_{sn} = \frac{\eta_{sn}\Omega_n}{K(\Omega_n)}sin\theta + \frac{\eta_{cn}\Omega_n}{K(\Omega_n)}cos\theta, \tag{17}$$

$$F_{cn} = -\frac{\eta_{cn}\Omega_n}{K(\Omega_n)}sin\theta - \frac{\eta_{cn}\Omega_n}{K(\Omega_n)}cos\theta. \tag{18}$$

So, once the Doppler shift variations are found, one can retrieve the local tilt of the surface, the incidence angles of radiation and the local values $R$, thus making it possible to obtain the local values of the non-Bragg component (Equation (4)).

## 3. Experiment

Experiments were conducted on an oceanographic platform located 500 m from the shore in the Black Sea (see Figure 2); the sea depth near the platform was about 30 m. An X-band scatterometer and a three-band microwave radar were located on the upper platform deck at a height of about 13 m above sea level, approximately 5 m apart. The directions of observation were approximately the same for the radars. The incidence angle was 55–60 degrees.

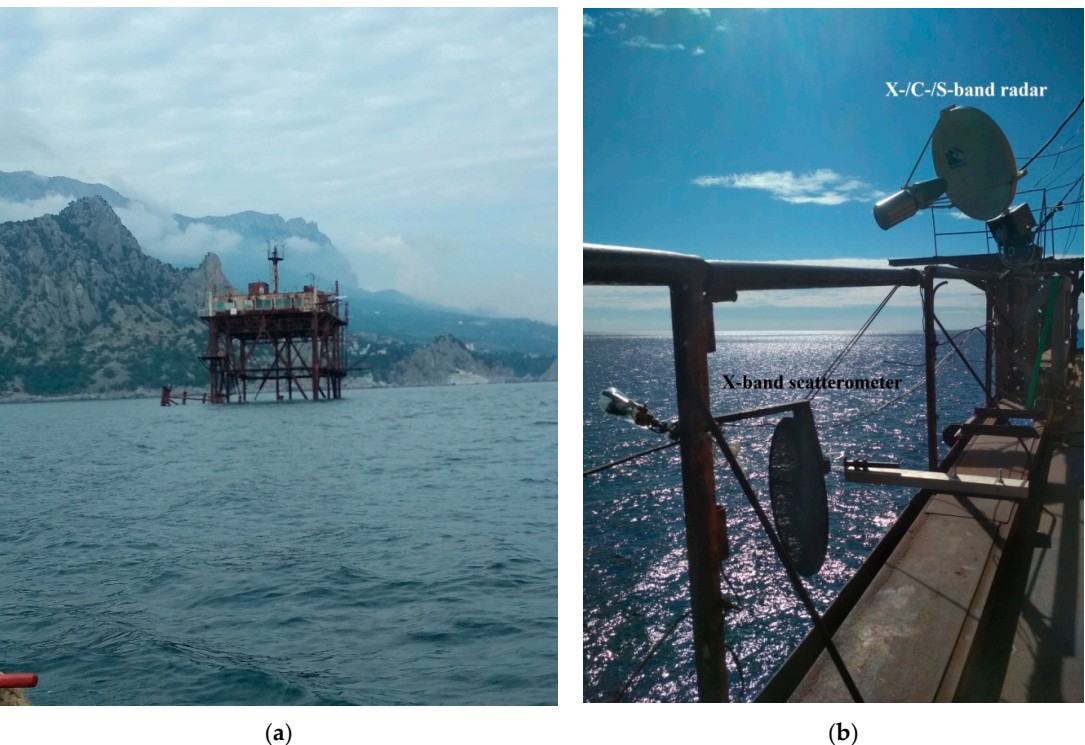

(**a**)　　　　　　　　　　　　　　　(**b**)

**Figure 2.** (**a**) Oceanographic platform and (**b**) the X-band scatterometer and the X-/C-/S-band radar mounted on the platform.

The X-/C-/S-band radar was a pulse system which simultaneously measured the intensity of radar return at three microwave frequencies and the corresponding speeds of radar scatterers. The scatterer speed was determined by the Doppler shift that was defined as the "centroid" of the Doppler spectrum. The radar was equipped with a parabolic 90 cm-diameter antenna. In our experiments, the sampling period was 4 s and the Doppler spectra of radar return were analyzed over time intervals of 0.5 s in each band. Since the sampling rate was comparable to the frequencies of long waves at moderate winds, the radar data were used to obtain mean radar return values. Since transmitted and received signals passed through the same electronic path, we could consider the amplification factors for the channels of different polarizations (VV and HH) as equal. This assumption was also confirmed by measuring signals after a mechanical rotation of the transmitter at the conditions of the constant wind velocity. The obtained accuracy of determination of polarization ratio related to limiting averaging time and noise limitations was about 10–15% in the three-band radar. The multiband radar was also used for normalization of the signal intensities in different channels of the X-band scatterometer, which was in fact a combination of two co-polarized scatterometers with a single antenna.

The X-band scatterometer operated in a continuous wave regime with linear frequency modulation. The sampling rate was 1000 Hz and the Doppler spectra were analyzed over time intervals of 0.25 s, which allowed us to study variations of radar return on the scale of dominant waves. The antenna beam width of the X-band scatterometer was about 6∘ and the footprint on the water surface had a size of 1.5 m (approximately the same as the three-frequency radar), which was more than ten times less than the wavelengths of the dominant waves at moderate velocities.

The wind velocity and direction were measured with a Wind Sonic acoustic anemometer mounted on the platform roof. Surface gravity waves were measured using a wire wave gauge located between radars under the platform. The near-surface current was measured with an acoustic current velocity profiler. Radar observations of the sea surface were undertaken for clean and contaminated water surfaces. In the experiments with surface-active films, solutions of oleic acid (OLE) in ethanol were poured onto the water surface from a small boat at a distance of about 100 m upwind from the platform, with results suggesting that the film near the platform was in a stationary state [26]. Surface concentrations of the surfactant films, roughly estimated as the spill volume divided by the slick area, were more than the saturated concentration of the oleic acid monolayer. Since in this case the dynamic film elasticity, which determines the short-wave damping, weakly depended on the mean film thickness, we supposed that the elasticity remained nearly constant during the experiments [27]. Video and photo recording of the surface state were carried out to monitor visually the state of the waves and to identify the zones of wave breaking. The areas of broken film in slicks were recorded visually and then excluded from the analysis.

## 4. Results

### 4.1. Mean Polarization Ratio

It was shown in [10] that, at moderate wind speeds in the presence of wave breaking, mean PR values weakly depend on wind velocity. Here we extended our observations and included cases of weak winds. Figure 3 shows the intensity of radar return at VV polarization, demonstrating rapid radar backscatter growth when wind speed exceeded values of about 1.5 m/s. One would expect that the wind ripples in very low-speed winds near the threshold of wave generation could be characterized as an ensemble of linear, small-amplitude gravity-capillary waves for which Bragg scattering is realized. According to Bragg scattering, at the incidence angle of 60° the polarization ratio would be about 27 for the X-band. However, the PR values near the threshold in Figure 3 were significantly smaller, just a few units, and increased with wind but did not exceed values of about 3–5 in the studied wind velocity range. One thus can conclude that the Bragg model is hardly applicable, not only for cases in which strong wave breaking is observed on the sea surface, but also for the weak wind conditions near the threshold of wind waves generation.

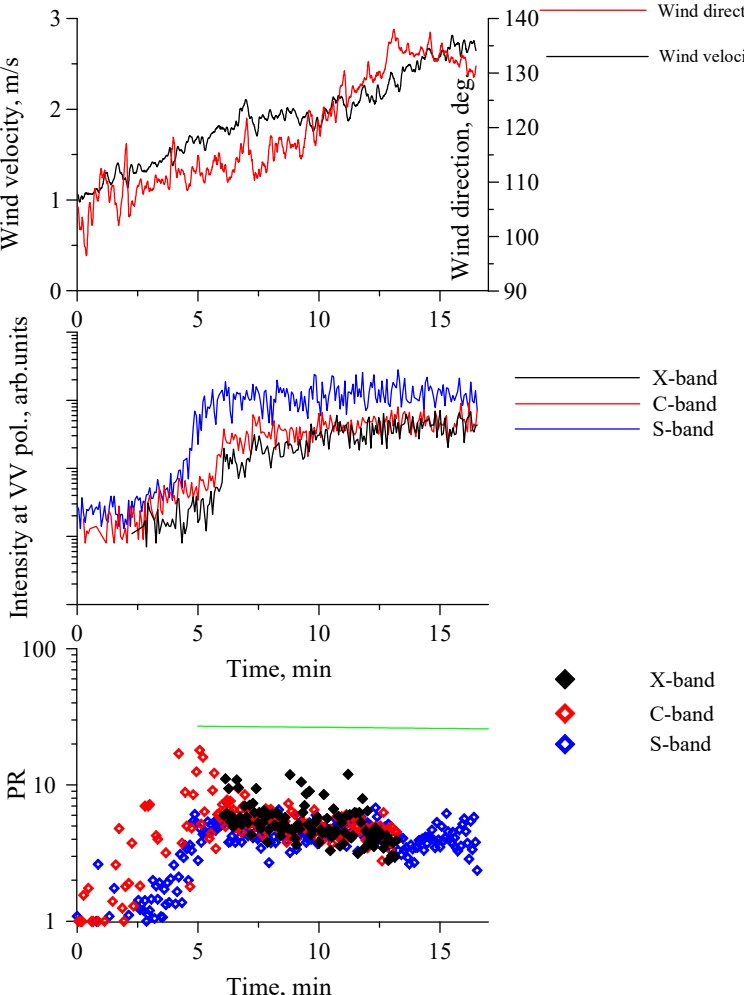

**Figure 3.** A case study of radar backscattering from the sea surface at growing wind velocity (top, black—wind velocity, red—direction), VV radar return (center, blue—C-band, red—S-band, black—X-band) and with polarization ratio (PR) variations (bottom, black symbols—X-band (multiband radar and X-band scatterometer), red symbols—S-band, blue—C-band); the ratio of $R(\theta)$ estimated according to the framework of the Bragg theory is denoted by the green line.

We have combined data of observations from a wide range of wind velocities. Figure 4 demonstrates the polarization ratio for wind speeds in the range of 4–12 m/s in the direction close to upwind (azimuth angles varied in the range of ±20° to the wind direction). The red crosses and black rhombuses denote $\langle PR \rangle$, averaged over a time interval of the order of several minutes, obtained using the X-band scatterometer and the multiband radar (X-channel) data, respectively; the red rhombuses mark the $\overline{PR}$. $\overline{PR}$ was close to $\langle PR \rangle$ in weak winds; that is, in the absence of long waves. A Significant difference between $\overline{PR}$ and $\langle PR \rangle$ appeared at moderate winds when rare but intensive spikes due to backscattering from breaking crests were observed in radar return. Figure 4 shows that the values of the PR in moderate winds depended on the data averaging procedures.

In order to illustrate the contribution of spikes to $\overline{PR}$ and $\langle PR \rangle$, we processed a two-minute recording of the X-band scatterometer signal at VV and HH polarizations (see Figure 5). One can see that the radar return at HH polarization contained a low-intensity "background" level and randomly distributed rare and strong spikes. Visually, these spikes were associated with strong breaking and white capping of the wave crests; the mean intervals between the spikes were several times larger than the typical long wave periods. In Figure 6a,b, the histograms of the instantaneous values of the radar return are plotted. The backscatter at HH polarization outside of spikes had small intensities

("HH background"). The values of the "background" at VV polarization were higher than those at HH polarization and comparable to the intensity of the spikes. In order to estimate the contribution of the spikes to radar backscattering, the total energy contained in a given intensity interval was also plotted in Figure 6 (see (c,d)). Figure 6 shows that the energy of the spikes was about 30% of the total backscattered energy, although the relative duration of the spikes was less than 2%. For VV polarization, the spike energy was less than the total energy of the "background". Figure 7 presents a scatter plot of the polarization ratio and the NBC (Figure 7a) and a histogram of the polarization ratio (Figure 7b) for the same case study. A "centroid" of the histogram was $\langle PR \rangle$. When retrieving $\langle PR \rangle$, the main contribution was made by the areas outside the spikes, as the duration of the spikes was small. To estimate the contribution of both spikes and non-spikes to the total radar return, the averaging should be performed with the weight of intensity of the NBC. Figure 7a shows that high values of NBC intensity corresponded to small values of local PR, so $\overline{PR}$ is less than $\langle PR \rangle$. In any case, both $\langle PR \rangle$ and $\overline{PR}$ were much smaller than the values in the Bragg model, thus indicating a significant share of the non-Bragg scattering in the total radar return. The relative contribution of non-Bragg scattering at low wind speeds was about 0.2–0.25 at VV polarization and about 0.75–0.8 at HH polarization.

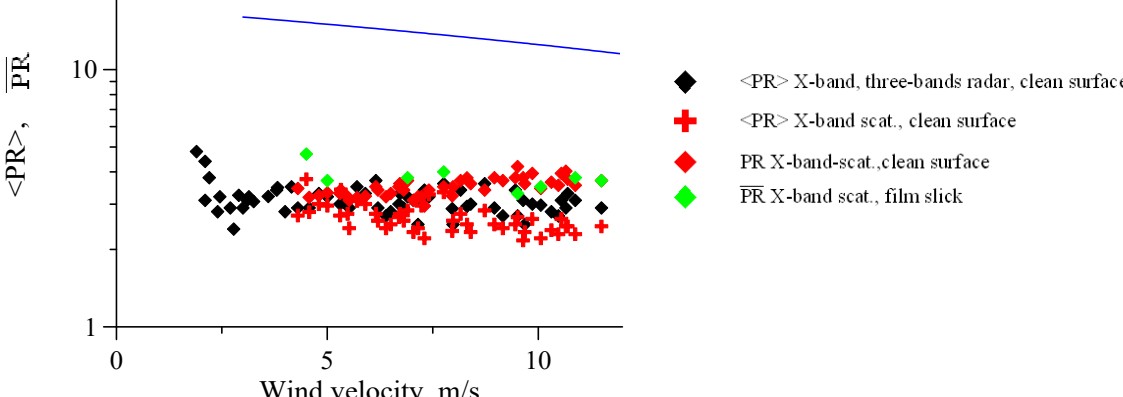

**Figure 4.** Mean PR vs. wind speed, upwind direction, X-band radar. Black symbols correspond to the three-band radar. Red rhombuses indicate the $\langle PR \rangle$ of the X-band scatterometer, red crosses indicate the $\overline{PR}$ of the X-band scatterometer, green rhombuses indicate the PR for slicks (the X-band scatterometer). The blue line corresponds to two-scale Bragg scattering theory. Incidence angles are 55–60 deg.

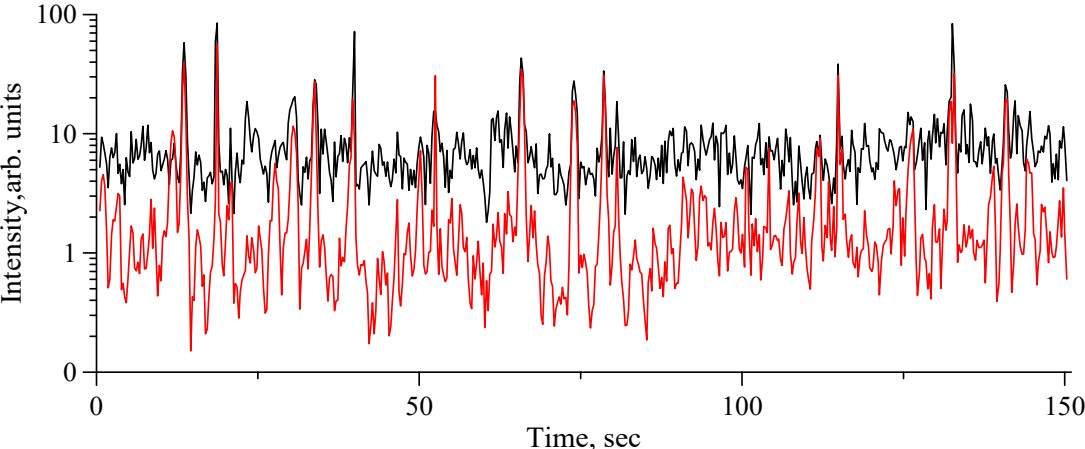

**Figure 5.** Record of radar return at VV (black curve) and HH (red curve) polarizations; wind velocity of 10 m/s, incidence angle of 55 degrees.

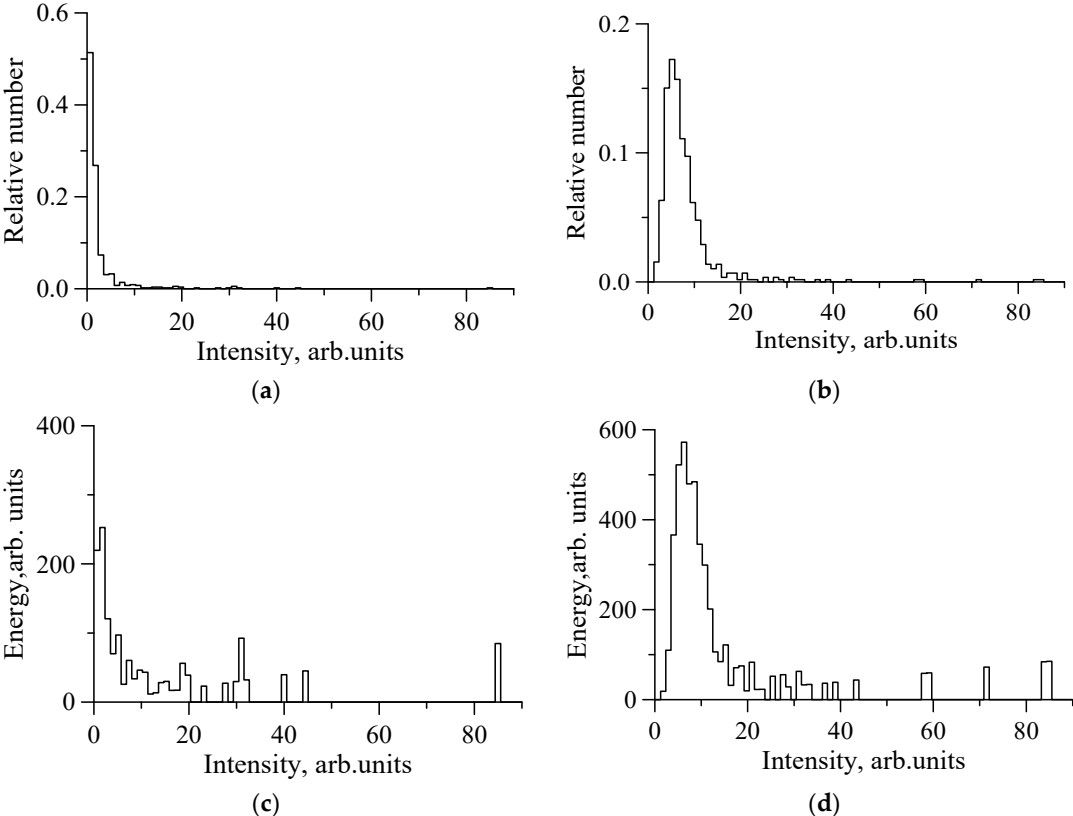

**Figure 6.** Histograms of instantaneous backscatter values (**a**,**b**) and diagrams of the total energy of local radar return within a given intensity interval (**c**,**d**) vs. intensity of local backscatter value at HH polarization (**a**,**c**) and VV polarization (**b**,**d**) for the case shown in Figure 5.

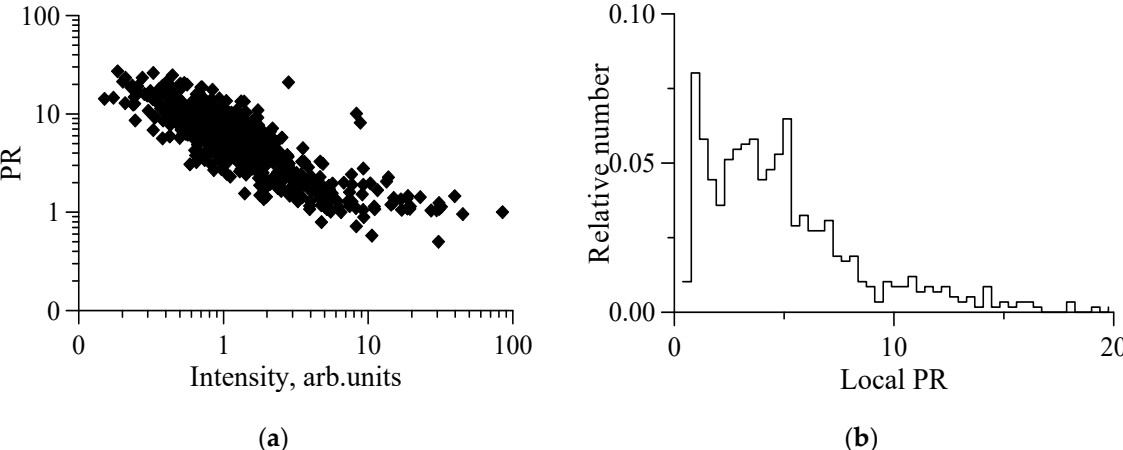

**Figure 7.** A scatterplot of polarization ratios vs. radar return intensity at HH polarization (**a**) and a histogram of polarization ratio values (**b**).

Figure 8 illustrates ⟨*PR*⟩ as a function of the azimuth angle for weak (4–5 m/s) (Figure 8a) and moderate (8–10 m/s) (Figure 8b) winds. The azimuth angle of 0° corresponds to the upwind observation. The dependences demonstrate that ⟨*PR*⟩ varied in the range of about 3 to 5 units for both cases, which was several times less than the polarization ratio for the Bragg theory.

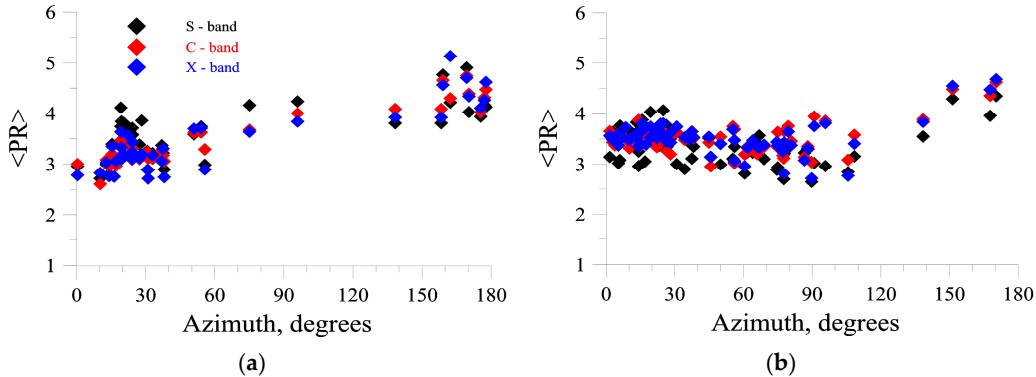

**Figure 8.** Polarization ratiovs. azimuth angle (0° corresponds to the upwind sensing direction). Weak winds (**a**), moderate winds (**b**). Black symbols—X-band, red—S-band, blue—C-band.

One can thus conclude that the relative contribution of the non-Bragg mechanism to the total backscattering is large and weakly depends on wind velocity for a wide range of wind speeds and azimuth angles.

### 4.2. Non-Bragg Component and Non-Bragg ScattererVelocity

When retrieving the non-Bragg component, we faced the problem of negative intensity values, which indicated some shortcomings of the model we used or which could be associated with measurement errors. Those cases can be neglected in further processing because of their rare occurrence. As mentioned above, the X-band scatterometer had a fine temporal/spatial resolution and allowed us to distinguish the wave-breaking areas and the "background" level of backscattering outside the spikes. An example of the retrieval of a non-Bragg component (a fragment of the record in Figure 5) using the procedure described in Section 2 is depicted in Figure 9. The figure demonstrates the Doppler shift of the Bragg component and the tilts of the long waves (a), the local polarization ratio in the Bragg model (Figure 9b) and non-Bragg components of radar return (Figure 9c). For comparison, in Figure 9a, the Doppler shifts of non-Bragg scatterers are given; it can be seen that non-Bragg scatterers both in and out of the wave-breaking areas moved with higher speeds than Bragg scatterers. It can also be seen that the slopes of long waves did not change the intensity of the spike, but changed the non-Bragg component outside the wave-breaking areas. Figure 10 presents a histogram of local values of the polarization ratio in the Bragg model for the case study presented in Figure 5. We can see that the histogram is shifted significantly towards larger values as compared to the histogram of the experimental polarization ratio (see Figure 8b) due to the contribution of non-Bragg scattering. Figure 11 shows the non-Bragg component energy diagram. We can see that the energy of the "background" was about half of the total energy of the non-Bragg component. The analysis carried out for case studies at wind speeds of 8–10 m/s showed that the "background" contributed 30–50% to the total energy of the NBC.

To estimate the current velocity, one can assume that Bragg scatterers move with the velocities of free waves, the intrinsic phase velocity of which is equal to:

$$V = \sqrt{\frac{g}{k_B} + \sigma k_B / \rho},$$ (19)

where $\sigma$ is the water surface tension and $\rho$ is the water density. To obtain the current velocity, one should subtract the intrinsic phase velocity of free waves from the total scatterer velocity, retrieved from Doppler shifts. However, we should note that the free wave assumption is a rather rough one since numerous experiments show that the wind wave spectrum also contains bound waves moving with the velocities of longer surface waves (see, for example, [28]). Therefore, we estimated the current velocity based on the wind speed and acoustic current velocity profiler data. The instantaneous

Doppler shifts of non-Bragg scatterers averaged over long surface waves (at high wind within and outside spikes) are shown in Figure 12. The corresponding wavelengths of the waves bound with the scatterers characterized by those Doppler shifts are also shown there.

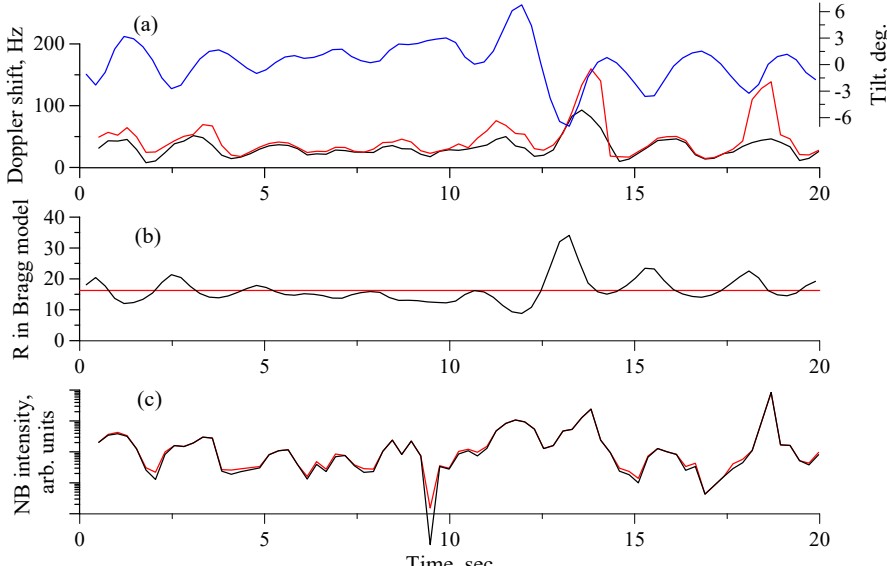

**Figure 9.** An example of a retrieval of a non-Bragg component showing: Doppler shift of the polarization difference (black curve), non-Bragg component (red) and tilts of the long waves (blue) (**a**); the local polarization ratio in a two-scale Bragg model (black) and mean PR (red) (**b**); and non-Bragg components of radar return when we did and did not take the local angle of the surface local tilts into account (**c**, black and red curves, respectively).

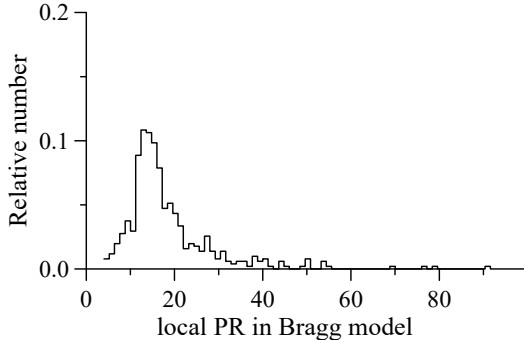

**Figure 10.** Histogram of local values of the PR in the Bragg model for the case shown in Figure 5.

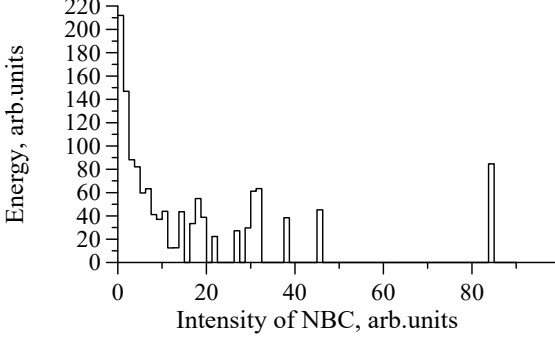

**Figure 11.** Energy of local non-Bragg component (NBC) contained in a given intensity range vs. NBC intensity for the case shown in Figure 5.

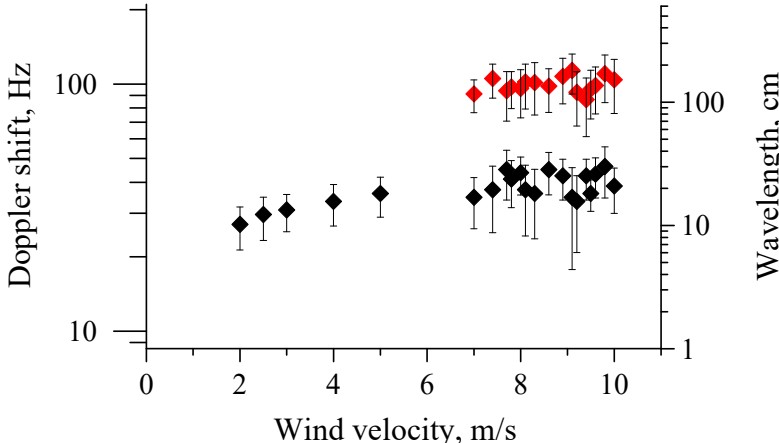

**Figure 12.** Doppler shifts in the background and in the spikes (black and red symbols, respectively) and the lengths of waves responsible for scattering. Confidence intervals of 95% are depicted for the wavelengths.

### 4.3. Non-Bragg Component in Film Slick

Now we analyze the role of surfactant films on the sea surface in non-Bragg backscattering. The experiments were conducted at moderate winds in the upwind direction (Table 1). Figure 13 shows contrasts (the ratio of radar returns for clean and contaminated sea surface) for the polarization difference and for the non-Bragg component outside spikes and contrasts in the spike areas. The difference between the mean velocities of Bragg and non-Bragg scatterers in non-slick and slick areas was estimated and also plotted in Figure 13. This figure shows that NBC intensity strongly decreased in film slicks and the NBC contrasts were close to the BC contrasts, while the spikes were weakly attenuated. The Doppler shifts of both BC and NBC components in slicks varied within the measurement error.

**Table 1.** Observation of film slicks.

| Data | Surfactant | Wind Velocity, Direction | Radar Azimuth | Comments |
|---|---|---|---|---|
| 29.05.2017 | OLE | 7.7; 170 | 180 | |
| 29.05.2017 | OLE | 9.8; 160 | 180 | |
| 28.05.2019 | OLE | 11.5; 170 | 160 | |
| 28.05.2019 | OLE | 9.5; 170 | 160 | No spikes in slick |
| 29.05.2019 | OLE | 10; 170 | 160 | |
| 29.05.2019 | OLE | 11.5; 175 | 160 | |

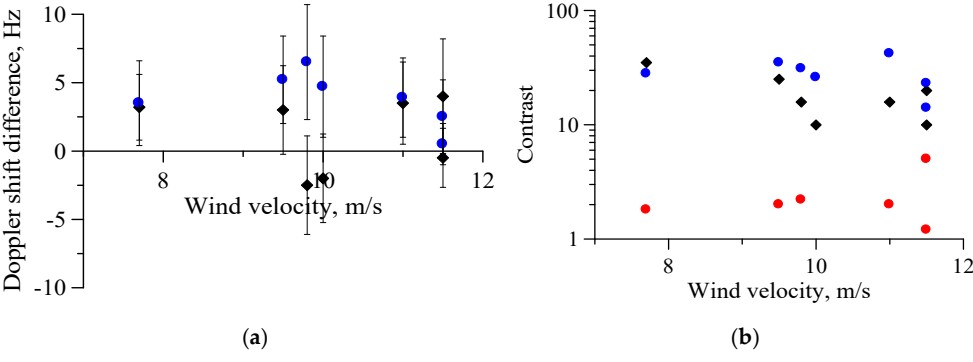

(**a**)          (**b**)

**Figure 13.** Doppler shift difference in slick and non-slick zones (**a**) and contrasts vs. wind speeds (**b**) for the X-band scatterometer and OLE slicks. Black symbols indicate the NBC "background" contrast and Doppler shift difference, blue dots indicate the polarization difference contrast and Doppler shift difference, red circles indicate the NBC contrast in spikes. Confidence intervals of 95% are depicted for Doppler shift difference.

## 5. Discussion

As shown in Section 4.1, the polarization ratio obtained in our experiments was less than the corresponding value in the Bragg model for all wind speeds and for all azimuth angles of our observations, which were nearly the same in C-, S- and X-bands. The significant contribution of the NBC to the total radar return at low winds and the fact that NBC scatterer velocities correspond to the wave velocities of dm waves (see Figure 12) allowed us to assume that the source of non-Bragg scattering was related to stable nonlinear structures "frozen" in the dm wave profile; that is, the structures attached to the dm-scale profile and moving steadily with the waves, for example, bulges and toes [19,21,22,29]. Strong wave breaking was an additional source of NBC that appeared at moderate winds and was comparable in energy with NBC from small-scale waves. The Doppler shifts of the spikes are shown in Figure 12; it can be seen that the wavelengths in spikes were larger and corresponded to m-scale surface waves. One can thus see that there are two types of non-Bragg scatterers associated with surface waves of dm- and m-scale wavelengths, respectively. The contributions of each of these types are comparable to each other at moderate wind speeds.

A qualitative explanation of weak dependence of $\langle PR \rangle$ on wind velocity is as following. Since the relative contribution of non-Bragg scattering is about 0.2–0.25 at VV polarization and about 0.75–0.8 at HH polarization, we can estimate the polarization ratio as a ratio between the Bragg component at VV polarization and the non-Bragg component:

$$\langle PR \rangle \approx \langle \frac{\sigma^0_{B\_VV}}{\sigma^0_{NBC}} \rangle \approx \frac{16\pi k_e^4 \langle g^2_{VV}(\theta)F(k_B) \rangle}{\langle \sigma^0_{NBC} \rangle}. \tag{20}$$

Since the non-Bragg component is associated with dm-scale waves, we assume that its intensity is proportional to the wind wave spectra at dm-scale wavelengths, i.e., $\sigma^0_{NBC} \sim F^S(k_{dm})$, where Sis some empirical factor. According to [5], the spectrum of a wind wave is:

$$F(k) = \alpha \left[ \beta(k, u_*^2) - 2 \cdot \gamma(k) \right]^{1/n} / k^4. \tag{21}$$

Here $\beta(u_*, k, \phi) \sim f(\phi)\frac{u_*^2}{\omega}k^2$ is the wind wave growth rate [30], $f(\phi)$ describes the dependence of the growth rate on the angle between the wind velocity and the wave vector $f(0) = 1$ in the upwind direction, $u_*$ is the wind friction velocity, $\omega = \omega(k)$ is the wave frequency, $\gamma(k) = 2\nu k^2$ is an amplitude viscous damping coefficient of surface waves, $\nu$ is the kinematic water viscosity and $\alpha$ is an empirical constant. It is assumed in Equation (21) that $\beta(k, u_*) > 2\gamma(k)$. For cm-scale waves, in particular, at $k = 3.6$ rad/cm corresponding to the X-band Bragg wave number for the conditions of our experiments, one can set $n = 1$ [15]. Then $n$ grows with wavelength and for dm-scale waves with wavelengths of 20 cm and longer can be considered to be about 5 [15]. At wind velocities higher that the threshold value, the polarization ratio is:

$$PR \sim \frac{\beta(k_{br}) - 2\gamma(k_{br})}{[\beta(k_{dm}) - 2\gamma(k_{dm})]^{S/n}}. \tag{22}$$

Here we have introduced an empirical factor S, which will be estimated below to fit with our experiment. If we can neglect the wave damping coefficient in Equation (22), which can be done at wind velocities of about 5 m/s and higher, then we obtain $PR \sim u_*^{2-2S/5}$. If the factor S~5, then the polarization ratio is independent of wind velocity; that is, consistent with the experiment (see Figure 4). When considering cases of film-covered surface we suppose that the reduction of a non-Bragg component occurs due to suppression of dm-scale waves. The nonlinear structures, responsible for non-Bragg scattering, are very sensitive to the amplitude of dm-scale waves and the dependence of the non-Bragg component on the spectral intensity of dm-scale waves should be very fast-growing (see, for example, [31]) like, e.g., a power function with high enough S values, so the

estimate of 5 for the *S*-value does not look unrealistic. The strong response of non-Bragg scattering to weak variation of dm-scale waves is consistent with the nonlinear "cascade" modulation of Ka-band radar backscatter due to internal waves [32]. One should remember that the dispersion relation for dm-scale waves is practically insensitive to the surface tension coefficient (STC), so that the reduction of STC in the presence of surfactant films does not change the phase velocity of dm-scale waves and thus the non-Bragg Doppler shift. Regarding the Bragg Doppler shift, the film should have a weak influence on cm-scale Bragg waves. If the non-Bragg component is associated with dm-scale waves and $\sigma^0_{NBC} \sim F^S(k_{dm})$, then the contrast (damping ratio) of the non-Bragg component is $C_{NBC} \sim C^S_{dm}$; here $C_{dm}$ is a contrast for dm-scale waves. The contrast of 2.3–3.2 Hz waves(which correspond to wavelengths of 15–30 cm) according to our wire gauge data for a wind velocity of 8 m/s is about 2. Thus, the non-Bragg contrast is about 30, which is consistent with the contrasts in Figure 13.

Note that a similar effect was observed in laboratory conditions in experiments with a Ka-band radar [19]. Specifically, the intensity of the non-Bragg component reflected from steep dm-scale waves decreased due to suppression of bulge/toe structures near the crests of the waves on the contaminated surface, while the velocity of dm-scale waves remained practically unchanged.

## 6. Conclusions

A significant difference between the polarization ratios predicted by the Bragg theory and those obtained in field experiments was found fora wide range of wind velocities, including cases of weak winds near the threshold of wind wave generation. This indicates that strong waves are not the only sources of non-polarized microwave scattering. The polarization ratio weakly depends on wind velocity and on the azimuth angle.

Analysis of Doppler shift of the radar spectra and scattering velocities revealed two types of non-Bragg scatterers. One type was associated with strong breaking and wave crest overturning; these scatterers moved with the phase velocities of meter-scale breaking surface waves. Another part of the non-Bragg scattering was due to microbreaking of dm-scale waves and formation of nonlinear structures, like bulges and toes, near the wave crests. The second type of non-Bragg scatterers dominated at low wind conditions, while at moderate wind velocities both strong breaking and microbreaking were responsible for non-Bragg microwave scattering. At moderate winds, the non-Bragg component related to microbreaking was approximately half of the total non-Bragg component.

The presence of surfactant films on the surface of the sea resulted in a significant suppression of both the small scale non-Bragg component and the Bragg component, so that the polarization ratio changed little. We believe that the nonlinear structures associated with dm-scale waves were strongly reduced in the presence of film due to the cascade mechanism, which results in strong suppression of small-scale nonlinear structures even if the reduction of the amplitude of the dm-wave is weak. At the same time the velocities of non-Bragg scatterers remained practically the same as in non-slick areas, since the phase velocity of dm waves was not affected by the film.

We believe that the presented results regarding the contributions of large breaking meter-scale waves, as well as microbreaking decimeter-scale waves, to non-polarized backscattering are valid, at least qualitatively, for a wide range ofradar acquisition geometry typical for satellite SAR both for upwind and waves directions. Moreover, this contribution can increase, taking into account that for smaller incidence angles for satellite SAR quasi-specular non-polarized scattering can be realized even for smaller slopes of the parasitic ripples and bulges.

**Author Contributions:** Conceptualization, I.A.S.; methodology I.A.S.; data curation, I.A.S. and O.V.S.; writing—original draft preparation, S.A.E.; writing—review & editing, I.A.S., O.V.S. and S.A.E.; investigation, A.V.E., I.A.K. and A.V.K. All authors have read and agreed to the published version of the manuscript.

**Funding:** This research was funded by the Russian Science Foundation (Project RSF 18-17-00224).

**Acknowledgments:** We are grateful to Alexander Molkov for his help in the experiments.

**Conflicts of Interest:** The authors declare no conflict of interest.

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
