# Peer review of "The Role of Micro Breaking of Small-Scale Wind Waves in Radar Backscattering from Sea Surface"

_remotesensing, doi:10.3390/rs12244159_

Round 1

Reviewer 1 Report

General Impressions:

This paper attempts to characterize the polarization effects in microwave radar due to nonlinearities of the ocean surface, e.g. breaking and near-breaking conditions. The authors seek to accomplish this though use of a polarization ratio within a linear polarization basis. The theoretical basis is tested on data taken by a collection of multi-band radars recently deployed on an oil platform in the Black Sea.

This is a well-written paper with sound arguments and a clear contribution to the remote sensing community; I enjoyed reading it. I have several questions and concerns enumerated in what follows, in no particular order. Each of my comments is minor.

Minor concerns:

  • Early on, the authors differentiate between polarimetric Bragg scattering and non-polarized, non-Bragg effects. I believe it should be emphasized in the beginning that the primary focus of the manuscript is on non-Bragg effects produced by local non-linearities in the ocean surface, rather than the large-scale tilting effects which are commonly described by the Kirchhoff Approximation, the Two-Scale Model, and others.
  • On page 6, line ~168, the authors state, “The accuracy of determination of polarization ratio is about 10-15% in the three-band radar.” It isn’t clear to me what “accuracy of determination” means in this context, or what the dominant error source is. Does this mean to say that the radar measurements are noise-limited, and that the polarization ratio accuracy is degraded by 10-15%?
  • Most of the plots herein would greatly benefit from the use of a legend.
  • In Figure 8 (and elsewhere) the y-axis labels should have “<PR>” rather than “PR” for the sake of clarity.
  • On page 11, line ~274 (and elsewhere), the authors refer to the “background” level of the radar measurements. Is this meant to refer to the noise floor? If so, this should be stated more explicitly.
  • Figures 12 and 13 show the confidence intervals of Doppler measurements, but it is unclear what percent confidence these represent -- 95%? 99%?
  • Page 14, line ~335 reads: “… the source of non-Bragg scattering is related to nonlinear structures ‘frozen’ in the dm wave profile….” It is unclear to be what “frozen” means in this context; this should be explained more clearly.
  • Equation 22 introduces the Donelan gravity wave spectrum, which forms the basis for a semi-empirical expression for the polarization ratio. As I understand it, the Donelan spectrum is most applicable to deep water waves, whereas the data analyzed here were taken in a littoral region. Some discussion or justification should be provided to explain the spectrum’s applicability in the littoral zone.

Spelling, grammar, and style:

  • Page 2, line ~65: “… correctly to interpret …” should be changed to “… correct to interpret …”.
  • Page 6, line ~182: “… clean and contaminated water surface …” should be changed to “… clean and contaminated water surfaces …”.
  • Page 9, line ~231: “… several time larger …” should be changed to “… several times larger …”.
  • On page 9, line ~240 (and elsewhere) the authors mention the so-called “center of gravity” of the histogram. I believe “centroid” is a better word here.
  • Page 11, line ~260: “… function of the azimuth …” should be changed to “… function of azimuth angle …”.
  • Page 11, line ~262: “… for both cases that is several times less that the polarization ratio for Bragg theory” should be changed to “… for both cases, which is several times less than the polarization ratio for Bragg theory”.
  • On page 13, line ~307: the acronym ADCP is not defined.
  • Page 15, line ~370: “One should remind …” should be changed to “One should remember …”

Page 15, line ~373: “… it also should be changed weakly for cm-scale Bragg waves” should be changed to “… the film should have a weak influ

Author Response

We are very thankful to anonymous reviewer 1 for comments. We made corrections, namely:

Comment 1: Early on, the authors differentiate between polarimetric Bragg scattering and non-polarized, non-Bragg effects. I believe it should be emphasized in the beginning that the primary focus of the manuscript is on non-Bragg effects produced by local non-linearities in the ocean surface, rather than the large-scale tilting effects which are commonly described by the Kirchhoff Approximation, the Two-Scale Model, and others.

Reply:  We emphasized in Introduction that the primary focus of the manuscript is on non-Bragg effects produced by local nonlinearities on the profile of short wind waves on the ocean surface (please see the new lines 80-83).

Comment 2: On page 6, line ~168, the authors state, “The accuracy of determination of polarization ratio is about 10-15% in the three-band radar.” It isn’t clear to me what “accuracy of determination” means in this context, or what the dominant error source is. Does this mean to say that the radar measurements are noise-limited, and that the polarization ratio accuracy is degraded by 10-15%?

Reply:  The polarization ratio obtained from the data of the three-band radar was estimated with accuracy of about 10-15% which was determined by variations of  VV-/HH-polrized radar return intensity due to limited averaging time and due to noise limitations in slick areas (please see lines 172-173)

Comment 3: Most of the plots herein would greatly benefit from the use of a legend.

Reply:   Figures 3, 4 and 8 are provided with legends.

Comment 4: In Figure 8 (and elsewhere) the y-axis labels should have “<PR>” rather than “PR” for the sake of clarity

Reply:   Corrected in Figures 3 and 8.

Comment 5: On page 11, line ~274 (and elsewhere), the authors refer to the “background” level of the radar measurements. Is this meant to refer to the noise floor? If so, this should be stated more explicitly.

Reply:   The backscattering signal outside the spikes (we called it “background”) is related to the non-Bragg scattering from the dm waves (lines 280-281).

Comment 6: Figures 12 and 13 show the confidence intervals of Doppler measurements, but it is unclear what percent confidence these represent -- 95%? 99%?

 Reply:   95% confidence intervals are depicted in the Figures. Changes have been made in the paper.

Comment 7: Page 14, line ~335 reads: “… the source of non-Bragg scattering is related to nonlinear structures ‘frozen’ in the dm wave profile….” It is unclear to be what “frozen” means in this context; this should be explained more clearly. 

Reply:   The term “frozen” structures is explained in the paper: The frozen structures are the structures attached to the dm-scale profile and moving steadily with the waves (please see the new lines 343-345)

Comment 8: Equation 22 introduces the Donelan gravity wave spectrum, which forms the basis for a semi-empirical expression for the polarization ratio. As I understand it, the Donelan spectrum is most applicable to deep water waves, whereas the data analyzed here were taken in a littoral region. Some discussion or justification should be provided to explain the spectrum’s applicability in the littoral zone.

Reply:   We have added the specification of the sea depth near the Platform where the experiments were conducted (~30 m, line 155). For dm waves these conditions can be obviously considered as a deep water case so we can use the Donelan gravity wave spectrum for our estimations.

Spelling, grammar, and style:

  1. Page 2, line ~65: “… correctly to interpret …” should be changed to “… correct to interpret …”.

Reply:    Corrected, new line 67.

  1. Page 6, line ~182: “… clean and contaminated water surface …” should be changed to “… clean and contaminated water surfaces …”.

Reply:   Corrected, new line 187.

  1. Page 9, line ~231: “… several time larger …” should be changed to “… several times larger …”.

Reply:   Corrected, new line 236.

  1. On page 9, line ~240 (and elsewhere) the authors mention the so-called “center of gravity” of the histogram. I believe “centroid” is a better word here.

Reply:    Corrected, new line 245.

  1. Page 11, line ~260: “… function of the azimuth …” should be changed to “… function of azimuth angle …”.

Reply:   Corrected, new line 265.

  1. Page 11, line ~262: “… for both cases that is several times less that the polarization ratio for Bragg theory” should be changed to “… for both cases, which is several times less than the polarization ratio for the Bragg theory”.

Reply:   Corrected, new lines 267-268.

  1. On page 13, line ~307: the acronym ADCP is not defined.

Reply:   ADCP is Acoustic Doppler Current Profiler. Corrected, new line 314.

  1. Page 15, line ~370: “One should remind …” should be changed to “One should remember …”

 Reply: Corrected, new line 379.

  1. Page 15, line ~373: “… it also should be changed weakly for cm-scale Bragg waves” should be changed to “… the film should have a weak influ

Reply:  Corrected, new lines 382-383.

Reviewer 2 Report

Dear author,

Please check the attached files.

Kind Regards.

Author Response

We are very thankful to anonymous reviewer 2 for comment.

MAJOR CONCERNS:

Comment 1:  I believe there are a few papers (out of which maybe only one is relevant) from Alexis Mouche discussing similar/comparable observation made from an airborne C-band radar which I think should be referenced to and put into context. (But no need to elaborate, a few lines or so should be enough).

 Reply:  Thank you for the valuable addition. We analyzed some papers and referred in Line 74 to Mouche, A., Hauser D., Daloze J.F. Guerin C. Dual-Polarization measurements at C-band over the ocean: results from airborne radar observation and comparison with Envisat ASAR data, IEEE Trans. Geosci. And. RS, 43,753769.

Comment 2:  Most of the operational oceanography relying in SAR remote sensing is done using satellite platforms for which the acquisition angles are within a different range. Can the authors say a few words about what is expected for these different ranges of incidence angles?

Reply: We believe that the presented results regarding the contribution of large breaking meter-scale waves, as well as micro-breaking decimeter-scale waves  to non-polarized backscattering are valid, at least qualitatively, in a wide range of radar acquisition geometry typical for satellite SAR both for upwind and waves directions. Moreover, this contribution can be ever larger taking into account that for smaller incidence angles [24] for satellite SAR the quasi-specular non-polarized scattering can be realized even for smaller slopes of the parasitic ripples and bulges (please see the new lines 415-420 in the Conclusions section). 

Comment 3: The reviewer acknowledges that there are previous papers by the authors that are important and precede the findings presented here. Nonetheless, a unifying schematics would greatly benefit this work, which could (for instance) show the radar acquisition geometry (upwind and waves directions), and the backscatter from large breaking meter-scale waves, as well as from micro-breaking decimeter-scale waves. It would also help considerably (within the larger communities of radar remote sensing and even physical oceanography), showing in this schematic view, what exactly are ‘toes’, ‘bulges’ and ‘parasitic capillary waves’.

Reply: We referred to Figure 1 of [Duncan, J.H. Spilling breakers. Annu. Rev. Fluid Mech. 200133, 519–547, doi:10.1146/annurev.fluid.33.1.519] where the nonlinearity schematics are presented (line 72).  In the near future, we hope to publish our new paper on laboratory research of nonlinearities on the profile of dm wave where photos and digitized wave profiles will be presented.   

MINOR CONCERNS:

The following lines are given for the new file:

  1. L. 38 - “a” is added
  2. L. 48 - “a” is added
  3. L. 52 – “interest in the problem”
  4. L. 62 – we specified the range of moderate and strong winds, in line 106 we added the range of low/weak winds.
  5. L. 181 – “ the” is added
  6. L.  195 – “to fix the zones” -> “ to identify the zones”
  7. L. 340 – “observation being nearly” -> “our observations which”
  8. Figure 3 and 4 – we use a log scale to demonstrate the significant difference between the Bragg-predicted and experimentally obtained values of PR. Figure 8 with a linear scale is focused only on the experimental results.

Reviewer 3 Report

Paper is good scientifically   however many edits needed

see attached

Author Response

We are very thankful to anonymous reviewer 3 for the comments. We have made corrections and listed the lines according to the new file:

  1. 14 – “supposed” -> “proposed”
  2. 20 – “is obtained” -> “was determined”
  3. Starting from L. 20 all “non polarized” was replaced by “non-polarised”, “non Bragg” was replaced by “non-Bragg”
  4. 37 – “speeds” -> “speed”
  5. 48 – the text is “At present, microwave radar, in particular the C- and X-bands, is a widely used remote sensing tool”
  6. 52 – “interest to” -> “interest in”
  7. 67 – “correctly” ->”correct”
  8. 70 – “high curvature values on the profile of short gravity waves” -> “high curvature values of the profile of short gravity waves”
  9. 81- the text was changed according to another reviewer’s comments, the corrections regarding replacing “data of” by ”data from” were also made
  10. 108 - “long waves slopes” -> “long wave slopes”
  11. 171 “in the conditions” -> “at the conditions”
  12. 184 – “mounted at” -> “mounted on”
  13. 214 – “multi- bands” -> “multi-band”
  14. 228 – “ three-bands” -> “three-band”
  15. 272 – “the” is added
  16. 328 – the text is “also plotted in Figure 13”
  17. 371 – “independent on” -> “independent of”
  18. 377 – “the” is added
  19. 379 – the text is “one should remember”
  20. 382 –“ it also should be changed weakly for cm- scale Bragg waves” -> “the film should have a weak influence on cm-scale Bragg waves”